# Effects of 3′-Sialyllactose on Symptom Improvement in Patients with Knee Osteoarthritis: A Randomized Pilot Study

**DOI:** 10.3390/nu16193410

**Published:** 2024-10-08

**Authors:** Eun-Jung Park, Li-La Kim, Hiroe Go, Sung-Hoon Kim

**Affiliations:** 1GeneChem Inc., Daejeon 34025, Republic of Korea; sksgy@genechem.co.kr (E.-J.P.); lilack8899@gmail.com (L.-L.K.); 2Department of Rehabilitation Medicine, Yonsei University Wonju College of Medicine, Wonju 26426, Republic of Korea

**Keywords:** 3′-sialyllactose, human milk oligosaccharide, osteoarthritis, knee osteoarthritis, visual analog scale, Western Ontario and McMaster Universities Osteoarthritis Index

## Abstract

**Background/Objectives:** 3′-Sialyllactose (3′-SL), a human milk oligosaccharide, has anti-inflammatory effects and is demonstrated to have protective effects against osteoarthritis (OA) in vitro and in vivo. However, this hypothesis remains to be investigated in a clinical setting. Herein, we investigated the effects of 3′-SL on pain and physical function in patients with knee OA. **Methods:** Sixty patients with knee OA with Kellgren and Lawrence grades (KL-grades) 1–4 and Korean Western Ontario and McMaster Universities Osteoarthritis Index (KWOMAC) scores ≥30 were randomly assigned to the placebo (*n* = 20), 3′-SL 200 mg (*n* = 20), and 3′-SL 600 mg (*n* = 20) groups. For 12 weeks, 3′-SL or placebo was administered to patients once a day. Clinical efficacy was evaluated using a visual analog scale (VAS) for pain and KWOMAC for physical function at baseline and at 6 and 12 weeks. Adverse effects were assessed for 12 weeks. **Results:** Significant reductions in VAS and KWOMAC scores were observed at 12 weeks compared with the baseline in the 3′-SL group. No severe adverse effects were observed over 12 weeks. **Conclusions:** 3′-SL reduced pain in patients with knee OA, improved daily life movements, and was safe, suggesting that 3′-SL might be an effective treatment for knee OA without severe side effects.

## 1. Introduction

Knee osteoarthritis (OA) is a multifactorial and degenerative whole joint disease that causes damage to the cartilage and surrounding tissues such as subchondral bone, meniscal, ligaments, capsule, and synovium [1]. It is characterized by the breakdown of the cartilage, subchondral bone remodeling, deterioration of tendons and ligaments, meniscal degeneration, inflammation, and fibrosis of both the infrapatellar fat pad and synovial membrane [1,2]. OA is the most common type of arthritis and is known to be caused by several risk factors such as musculoskeletal aging, obesity, joint injury, anatomical factors, and excessive joint use [3]. As the damage of cartilage and joint tissue including that of the infrapatellar fat pad and the adjacent synovial membrane progresses, several symptoms of OA such as pain, stiffness, swelling near the joint, loss of motion, and subsequent functional disability occurs [4]. These symptoms limit several activities in daily life and directly affect the quality of life, resulting in social and economic burdens [5]. In 2020, 595 million people worldwide—equivalent to 7.6% of the global population—had OA, an increase of 132.2% since 1990 [6]. Therefore, exploring effective treatments for OA is necessary.

Several interventions have been introduced to relieve OA symptoms, including lifestyle changes, pain relief medicines, and surgery; platelet-rich plasma, glucosamine, chondroitin, non-steroidal anti-inflammatory drugs, acetaminophen, and opioids have also been used [7,8,9,10,11,12]. However, data regarding the safety of the long-term use of these interventions and strong evidence of their treatment efficacy are lacking [13,14].

3′-Sialyllactose (3′-SL), a human milk oligosaccharide, is involved in several processes including immune activity, autoimmunity activity, brain development, cognitive enhancement, and anti-arthritic effects [15]. Inflammatory cytokine production and metalloproteinase (MMP)-induced collagen degradation are associated with OA pathogenesis [16]. Several studies have shown that 3′-SL improves OA symptoms in vitro and in vivo (rat and minipig) by inhibiting MMP production, inflammatory cytokine production, apoptosis, and oxidative stress [17,18,19]. Moreover, some in vitro, in vivo, and human pilot studies have demonstrated the safety of 3′-SL [20,21,22]. These findings suggest that 3′-SL is a feasible treatment option for OA symptoms; however, to the best of our knowledge, it has not been tested in human patients with knee OA. Therefore, we conducted a pilot clinical trial to analyze the safety of 3′-SL and its effect on pain and physical function in patients with knee OA.

## 2. Materials and Methods

### 2.1. Study Design and Subject Assignment

This randomized, single-blind, prospective three-arm study spanning 12 weeks was conducted on patients with knee OA at the Wonju Severance Christian Hospital in Wonju, Republic of Korea (between August 2019 and March 2020). Before the clinical trial, the Korean Western Ontario and McMaster Universities Osteoarthritis Index (KWOMAC) [23] was assessed, and patients with KWOMAC scores ≥30 were screened and randomized into experimental group 1 or 2, or the control group. For patients who met all exclusion criteria, random assignment numbers were randomly assigned to the experimental group or control group in a 1:1 ratio in the order registered by the investigator. A random assignment random number table is generated by a statistician independent of this clinical trial using statistical software (Version 9.4, SAS^®^ Institute, Cary, NC, USA) using block randomization, and a random assignment number of sufficient size is issued considering the pre-designated block size. A random assignment envelope is made with the issued random assignment number, and the random assignment envelope is delivered to the unblind investigator of the institution before subject registration. Each group consisted of 20 participants who were randomly assigned to receive either 200 mg 3′-SL (experimental group 1), 600 mg 3′-SL (experimental group 2), or a placebo (control group). Dosages were selected based on in vivo studies in mouse and minipig OA models [18,19]. Participants received one oral dose of 3′-SL or placebo daily for 12 weeks after the screening visit. Patients visited the study site at 6 and 12 weeks for functionality and safety assessments. To determine independent effects, the participants were instructed to participate in the clinical trial, maintaining the same conditions as before the trial, as smoking, exercise, diet, and cholesterol medication during the test period were reported to affect the study.

### 2.2. Ethics Statement

The purpose and content of this study were explained in detail to the participants or their guardians (legal representatives) and written consent was obtained. As part of the informed consent process, the patient or their legal representative was allowed to ask the investigator any questions regarding the risks and potential benefits of participating in the study. This study was approved (protocol code #CR319030 and date of approval 28 May 2019) by an independent Institutional Review Board and was conducted in accordance with the Korean Good Clinical Practice Guidelines and the principles of the Declaration of Helsinki.

### 2.3. Participants

The inclusion criteria were as follows: (1) Korean men and women aged over 40 years; (2) a total KWOMAC score ≥ 30 points; (3) Kellgren and Lawrence (KL) grades 1–4 in the OA radiographic classification system based on radiographs of the patellofemoral or tibiofemoral bones; (4) those with clinical and radiographic evidence of primary knee OA according to the American College of Rheumatology diagnostic criteria, meeting at least one of the clinical (morning stiffness in and around the joints lasting <30 min, and crepitus when the affected joint is moved) and radiologic symptoms (osteophyte formation on radiographs) [24]; (5) patients who voluntarily agreed to participate in the clinical trial and provided written informed consent; (6) those who were capable of understanding and following instructions related to the clinical trial; and (7) those who agreed to stop treatment and therapy for knee arthrosis, such as physiotherapy and injection therapy, during the clinical trial period.

The exclusion criteria were as follows: (1) more severe pain in other body parts than in the knee; (2) patients who continually received oral and injectable psychotropic medications and narcotic analgesics for the treatment of knee OA within 1 month of screening; (3) those taking OA-related medications; (4) patients with a history of lower extremity fracture or knee surgery (artificial joint replacement, metal fixation, etc.), or who needed to undergo knee surgery during the clinical trial period; (5) those with endocrine abnormalities (diabetes), other arthritis (rheumatoid and systemic lupus erythematosus), and gout (including pseudogout); (6) patients with orthopedic diseases that may affect functional assessment; (7) those with a history of peripheral vascular disease, varicose veins, deep vein thrombosis, or a risk of venous thrombosis (swelling of veins with pain and fever); (8) women who were pregnant or lactating, or those who may have become pregnant during the clinical trial and did not consent to the use of medically acceptable methods of contraception during the clinical trial period; (9) allergic reaction to the investigational product; (10) patients with a history of or currently having kidney-related diseases; and (11) patients with clinical findings that the principal investigator deemed inappropriate for the clinical trial.

### 2.4. Investigational Product

We used 3′-SL and placebo tablets as investigational products. Both tablets were identical in appearance, color coating, shape, and size. 3′-SL tablets contained 3′-SL and an excipient (microcrystalline cellulose) and were manufactured by GeneChem Inc. (Deajeon, Republic of Korea). Placebo tablets contained the same excipient as the 3′-SL tablets. All investigational products were provided in sachets and carefully stored at room temperature under dry conditions until distributed to the participants. 3′-SL was produced using the one pot reaction system invented by GeneChem and was dissolved in water. Briefly, substrates and enzymes (cytidylate kinase (CMK), polyphospate kinase (PPK), CMP-NeuAc synthetase (NEU), *N*-acetyl-D-glucosamine-2-epimerase (NANE), NeuAc aldolase (NAN), and α2,3-sialyltranferase (α2,3STN)) were mixed and reacted in a one-pot reactor.

### 2.5. Outcome Measures

The primary outcome measure was analyzed using the difference in the score of pain using the visual analog scale (VAS). The VAS is a pain scale that can be used as a questionnaire and provides a score ranging from 0 to 100, with higher scores indicating greater pain intensity [25]. This assessment tool was used at baseline and at 6 and 12 weeks, and the participants were asked to indicate the pain felt in the knee in the last 24 h after taking the tablets by drawing an exact vertical line on a parallel line.

The secondary outcome measure was analyzed using the difference in the KWOMAC total score and KL grade score. The KWOMAC is widely used in the evaluation of hip and knee OA. It is a self-administered questionnaire consisting of 24 items divided into three subscales (five for pain, two for stiffness, and seventeen for physical function). The scores for each subscale are summed, with a possible score range of 0–20 for pain, 0–8 for stiffness, and 0–68 for physical function; higher WOMAC scores indicate worse pain, stiffness, and functional limitations [26]. This assessment tool was used at baseline and at 6 and 12 weeks.

Information on adverse reactions was obtained from voluntary reports from the participants; additionally, data from participant diaries, investigator interviews, and medical examinations conducted during the clinical period were collected. The investigation of adverse reactions included the dates of onset and disappearance, extent and results of adverse reactions, causal relationship with the investigational product, and whether and what treatment was administered for the adverse reactions.

### 2.6. Statistical Analysis

All statistical analyses were performed using the GraphPad Prism 10.1.0 software (GraphPad Software, Inc., San Diego, CA, USA). All measurements in this study are presented as mean ± standard deviation and median (min–max) for continuous variables and frequency (percentage) for categorical variables. A one-way analysis of variance (ANOVA) or two-way ANOVA followed by Tukey’s multiple comparison test was performed to identify general character differences among groups and time points. Statistical significance was set at *p* < 0.05. The per protocol (PP) set included all participants who completed the study. The safety set included 60 subjects who had received at least one application of investigational products for a clinical trial. This study is a pilot clinical study and a formal statistical power analysis to determine the study sample size was not completed.

## 3. Results

### 3.1. Patient Characteristics

Herein, 60 patients with knee OA were enrolled, of whom 56 completed the study. Two patients on 600 mg 3′-SL were lost to follow-up by 6 weeks and two patients on placebo were lost to follow-up by 12 weeks—three without efficacy evaluation and one at the discretion of the investigator. The baseline characteristics of the included patients are shown in Table 1. The male-to-female ratio in each group was identical. The targeted side (right or left) of the knee joint in each patient was similar between the groups. Additionally, there were no significant differences in age, height, weight, and body mass index between the groups. Only the 600 mg 3′-SL group included patients with KL grades of 3 or 4.

### 3.2. Adverse Events

No serious adverse events were reported in any of the groups (Table 2). The observed adverse events were predominantly minor gastrointestinal, dermatological, and other miscellaneous symptoms. The most common adverse effects included nausea, indigestion, and heartburn. In the 60 participants, 12 adverse events were identified, all of which were mild in severity. These adverse events resolved completely after 12 weeks of treatment. Based on this, it was concluded that the investigational product used herein was well tolerated over the 12-week study period.

### 3.3. Clinical Outcomes

#### 3.3.1. Primary Outcome

The VAS pain scores were compared among the groups over a 12-week follow-up period (Table 3 and Figure 1). In the placebo group, no significant differences were observed in VAS scores over the 12 weeks. Conversely, both in the 200 mg and 600 mg 3′-SL groups, VAS pain scores significantly decreased at 12 weeks (200 mg: 18.00 ± 11.05, *p* < 0.01; 600 mg: 22.78 ± 17.76, *p* < 0.05) compared with the baseline values (200 mg: 36.00 ± 16.98; 600 mg: 40.50 ± 17.61) (Table 3 and Figure 1A). No significant differences were observed at 6 weeks in both the 200 mg and 600 mg groups. Additionally, no significant differences were observed in VAS scores among the groups at each time point (Table 3 and Figure 1B).

#### 3.3.2. Secondary Outcome

The KWOMAC scores decreased significantly after 12 weeks of 3′-SL treatment (Table 4 and Figure 2). The mean KWOMAC scores of the placebo group did not show a significant decrease (Table 4 and Figure 2A). However, the mean KWOMAC scores of the 200 mg 3′-SL group were 44.55 ± 8.73, 37.45 ± 9.97, and 34.55 ± 12.80 at baseline, 6 weeks, and 12 weeks, respectively, showing a significant difference only at 12 weeks compared with the baseline values (Table 4 and Figure 2A). In the 600 mg 3′-SL group, the mean KWOMAC scores were 44.8 ± 14.28, 34.50 ± 10.69, and 31.17 ± 7.76 at baseline, 6 weeks, and 12 weeks, respectively. Compared with the baseline values, the KWOMAC scores of the 600 mg 3′-SL group at 12 weeks significantly decreased (Table 4 and Figure 2A). No significant differences were observed in KWOMAC scores among the groups at each time point as in the VAS results (Table 4 and Figure 2B).

## 4. Discussion

This study assessed the safety and efficacy of 3′-SL in managing pain and enhancing physical function among patients with knee OA. Significant reductions in VAS and KWOMAC scores, which are the most widely used self-administrated scales to evaluate pain in hip or knee OA [27], following 3′-SL administration. The VAS measures joint-specific pain, and the KWOMAC reflects pain, stiffness, and function in hip and knee OA [28]. These symptoms serve as key treatment indicators for OA [29].

Regarding VAS findings, both 200 mg and 600 mg 3′-SL ameliorated pain only at 12 weeks. However, 200 mg 3′-SL (*p* < 0.01) showed a significantly greater improvement than 600 mg 3′-SL (*p* < 0.05). Patients in the 200 mg 3′-SL group had mild OA (KL grade 1 to 2), whereas the 600 mg 3′-SL group included four and two patients with moderate and high severity OA (KL grade 3–4), respectively. The difference in KL grades may have influenced the VAS results. No significant differences were observed in the VAS scores of the three groups at baseline, 6 weeks, and 12 weeks. Some studies have reported that placebos are considerably effective, and high placebo responses hamper the ability to identify new and effective treatments [29,30,31]. Our study also demonstrated a placebo effect over 12 weeks; therefore, no significant differences were observed among the groups at 6 and 12 weeks. Based on these observations, 200 mg 3′-SL may be more sufficient to relieve pain.

The KWOMAC scores were measured to assess the effect of 3′-SL on physical function in patients with knee OA. 3′-SL significantly reduced the KWOMAC scores at only 12 weeks in both the 200 mg and 600 mg 3′-SL groups. In contrast to VAS, 600 mg 3′-SL (*p* < 0.01) showed a significantly greater improvement than 200 mg 3′-SL (*p* < 0.05). In relation to physical function, 600 mg 3′-SL administration was more effective than 200 mg 3′-SL.

Overall, 3′-SL may be reliably effective in improving symptoms (pain and physical function) in patients with knee OA, even after considering the 12-week placebo effect. At least 12 weeks of 3′-SL treatment may be required to relive pain and improve difficulties in daily activities in patients with OA. We used 200 mg 3′-SL as a low dose and 600 mg 3′-SL as a high dose. We wanted to determine the difference in efficacy due to the difference in concentration, but we could not confirm the significant difference within these concentration ranges (200 mg–600 mg). It is not clear whether this was because the substance was not concentration-dependent or because the number of patients was small, making it difficult to clearly confirm the difference. However, the number of patients would need to be increased to confirm it again in the future.

As no cure is currently available for OA, long-term treatment is essential for OA prevention and mitigation, and thus, the safety of interventions should be ensured. Several types of natural products have been studied and used to promote joint health as an alternative to drug therapies such as NSAID therapy, which can cause a variety of side effects [32,33]. For example, treatment using krill oil, astaxanthin, and an oral hyaluronic acid complex for 12 weeks relieved symptoms of patients with mild OA [32]. Treatment using *Boswellia serrata*, ginger extracts, and *Harpagophytum procumbens* reduced symptoms of knee OA in clinical trials [34]. According to the network meta-analysis of treatments for knee OA through the PubMed, Embase, and Cochrane Library electronic databases from inception to October 2018, hyaluronic acid combined with platelet-rich plasma showed efficiency in improving physiological function and stiffness, as well as total WOMAC score, while platelet-rich plasma reduced pain according to WOMAC score [35]. Another network meta-analysis of intervention for OA by Cochrane reviews showed diet/weight loss and surgery has less effect than corticosteroids, herbs, mind and body exercises, orthotics, passive treatment, and regenerative medicine on pain [36].

3′-SL is reportedly safe and is used as a dietary supplement [37]. Several toxicological studies in rats, beagle dogs, neonatal piglets, and humans have demonstrated that 3′-SL is safe for human consumption. Briefly, the no-observed-adverse-effect-level of 3′-SL was >2000 mg/kg in a 13-week oral toxicity study of male and female rats [21]. Furthermore, the mean lethal dose of 3′-SL was >20 g/kg, as determined in a 14-day acute oral toxicity study of male and female rats [21]. In neonatal piglets, 500 mg/L 3′-SL did not affect the serum chemistry or microscopic structure of organs [22]. Human clinical studies have also observed that 3′-SL is well tolerated, with no major adverse effects up to doses of 20 g/day [20]. In this study, no serious adverse effects of 200 and 600 mg 3′-SL were observed over 12 weeks, suggesting that 3′-SL is an optimal long-term oral treatment for OA.

A weakness of this study is that it was a pilot clinical trial rather than a main study; therefore, the number of patients was small. A further study with a larger number of patients will be needed to ensure the effectiveness of 3′-SL on OA. As the number of patients increases, the difference in effectiveness according to concentration will also become clear. Moreover, only Korean participants were included in this study. For global use, it will be necessary to examine the effects on other populations in the future.

A limitation of this study is that the direct effects of 3′-SL on the knee were not examined. The direct effects of 3′-SL on the recovery of the cartilage structure should be revealed through imaging examinations, including magnetic resonance imaging, computed tomography, and ultrasound.

Although the exact mechanism remains unclear, OA has been associated with multifactorial events such as inflammation, MMP-induced collagen degradation, hyperactivated catabolic activity, oxidative stress responses, joint fibrosis, and biomechanical changes [16,17,18,32,38,39]. Studies have evaluated the mechanism of 3′-SL in in vitro and in vivo OA models and showed that 3′-SL inhibits cartilage degradation, inflammation, oxidative stress, cell apoptosis, and promoted cartilage regeneration [17,18,19]. These mechanisms of 3′-SL may be influenced to relieve pain and enhance physical function. Moreover, it would be worthwhile to examine whether 3′-SL inhibits the excessive secretion and deposition of extracellular matrix proteins that induce excessive fibrosis. Tissue fibrosis induces joint stiffness and pain in OA [38]. It is known that OA is a disease of the whole joint, and biomechanical changes also affect the pathogenesis and progression of OA [39]. The effects of 3′-SL on biomechanical tissue characteristics such as muscle atrophy, bone density, and subchondral bone sclerosis should also be determined. In the future, a detailed evaluation of these mechanisms should be conducted in patients with knee OA.

Since in vivo studies of 3′-SL on OA (mouse and minipig) used oral ingestion to treat 3′-SL, the oral route of administration was chosen for this pilot clinical trial. However, topical/transdermal administration, along with oral administration, are the most widely used routes of administration in the treatment of OA [40]. Oral administration has been reported to cause problems such as low bioavailability; perforation of the digestive tract, liver, and kidneys; and gastrointestinal injury [40]. Moreover, intra-articular-injected 3′-SL in a minipig model of rheumatoid arthritis was shown to have a therapeutic effect [41]. Administration injections would also be useful in the future to test the more efficient and rapid effects of 3′-SL.

## 5. Conclusions

In conclusion, the study findings suggest that 3′-SL is effective in reducing pain and improving daily life movements in patients with knee OA and is used as a safe and effective new treatment for knee OA. Moreover, this study is the first investigation of the effect of 3′-SL on human OA symptoms; it is valuable and can be considered a new discovery in this field.

## Figures and Tables

**Figure 1 nutrients-16-03410-f001:**
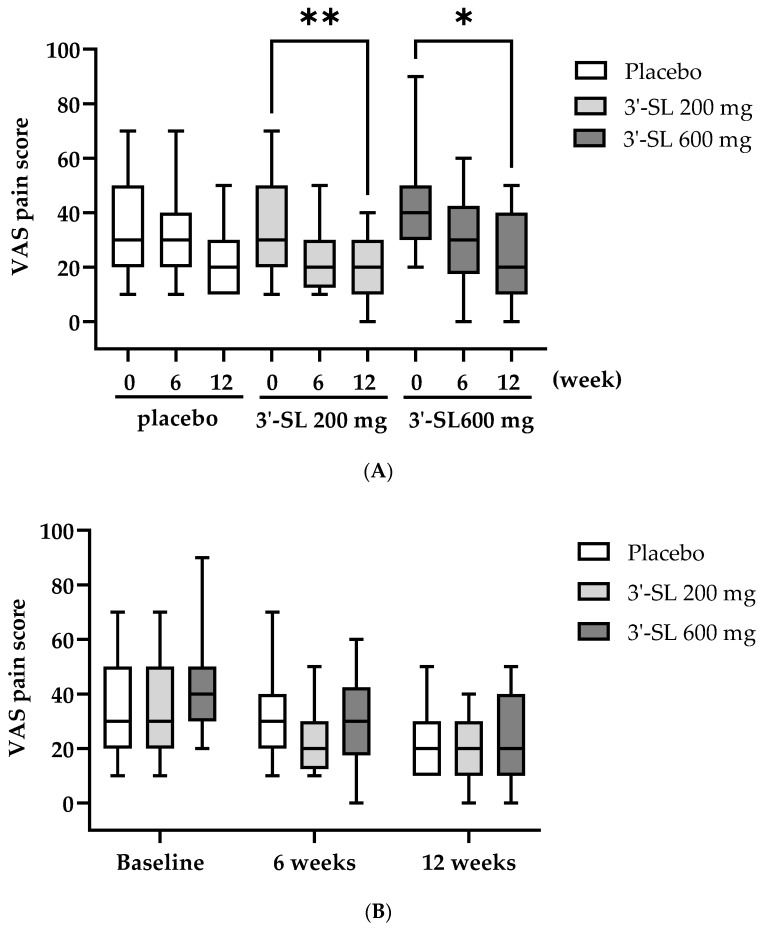
Changes in the VAS pain scores according to the group and time. The box and whisker plot (min to max) of the VAS pain scores based on the (**A**) group (placebo, 200 mg 3′-SL, and 600 mg 6′-SL) and (**B**) time (baseline, 6 weeks, and 12 weeks). At baseline: placebo, *n* = 20; 200 mg 3′-SL, *n* = 20; and 600 mg 6′-SL, *n* = 20. At 6 weeks: placebo, *n* = 20; 200 mg 3′-SL, *n* = 20; and 600 mg 6′-SL, *n* = 18. At 12 weeks: placebo, *n* = 20; 200 mg 3′-SL, *n* = 18; and 600 mg 6′-SL, *n* = 18. Significance was measured using a two-way analysis of variance followed by Tukey’s multiple comparisons test, * *p* < 0.05 and ** *p* < 0.01 vs. baseline. Abbreviations: VAS, visual analog scale; 3′-SL, 3′-sialyllactose.

**Figure 2 nutrients-16-03410-f002:**
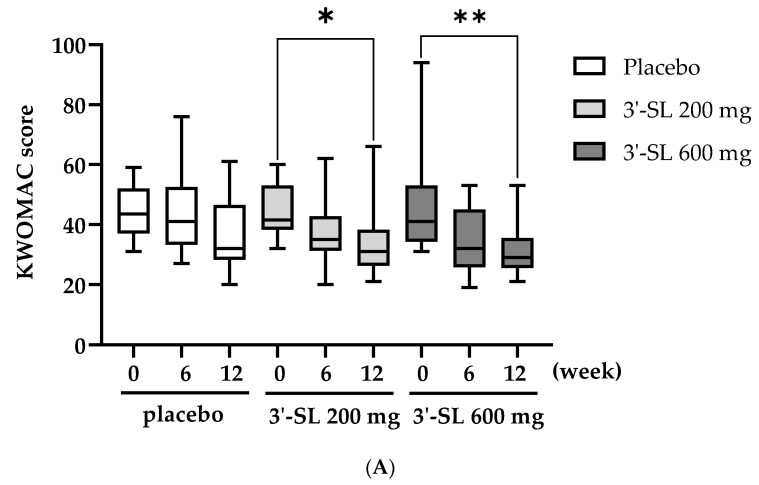
Changes in the KWOMAC scores according to the group and time. The box and whisker plot (min to max) of the KWOMAC quality of life questionnaire score based on the (**A**) group (placebo, 200 mg 3′-SL, and 600 mg 6′-SL) and (**B**) time (baseline, 6 weeks, and 12 weeks). At baseline: placebo, *n* = 20; 200 mg 3′-SL, *n* = 20; and 600 mg 6′-SL, *n* = 20. At 6 weeks: placebo, *n* = 20; 200 mg 3′-SL, *n* = 20; and 600 mg 6′-SL, *n* = 18. At 12 weeks: placebo, *n* = 20; 200 mg 3′-SL, *n* = 18; and 600 mg 6′-SL, *n* = 18. Significance was measured using a two-way analysis of variance followed by Tukey’s multiple comparisons test, * *p* < 0.05 and ** *p* < 0.01 vs. baseline. Abbreviations: KWOMAC, the Korean Western Ontario and McMaster Universities Osteoarthritis Index; 3′-SL, 3′-sialyllactose.

**Table 1 nutrients-16-03410-t001:** Characteristics of the study population.

	Placebo (*n* = 20)	200 mg 3′-SL (*n* = 20)	600 mg 3′-SL (*n* = 20)	*p*-Value
Age (years)				
Mean ± SD Median (min–max)	53.30 ± 5.65 53.50 (43.0–66.0)	53.55 ± 6.47 52.50 (45.0–71.0)	56.9 5 ± 10.18 54.00 (43.0–82.0)	0.2534
Height (cm)				
Mean ± SD Median (min–max)	1.64 ± 0.10 1.63 (1.45–1.86)	1.62 ± 0.07 1.64 (1.47–1.72)	1.63 ± 0.09 1.64 (1.43–1.79)	0.8388
Weight (kg)				
Mean ± SD Median (min–max)	67.78 ± 11.48 66.20 (49.0–92.2)	64.54 ± 8.97 63.05 (51.7–85.8)	66.35 ± 12.21 65.30 (54.0–106.9)	0.6484
Sex				
Male	10 (50.0%)	10 (50.0%)	10 (50.0%)	
Female	10 (50.0%)	10 (50.0%)	10 (50.0%)	
Targeted side of the knee joint				
Right	13 (65.0%)	9 (45.0%)	9 (45.0%)	
Left	7 (35.0%)	11 (I 55.0%)	11 (55.0%)	
BMI (Kg/m^2^)				
Mean ± SD Median (min–max)	25.23 ± 2.69 24.94 (20.26–32.11)	24.60 ± 2.81 24.37 (19.96–30.58)	24.91 ± 2.86 24.82 (20.25–33.25)	0.7786
KL grade	Grade 1: 4 (20.0%) Grade 2: 16 (80.0%) Grade 3: 0 (0.0%) Grade 4: 0 (0.0%)	Grade 1: 3 (15.0%) Grade 2: 17 (85.0%) Grade 3: 0 (0.0%) Grade 4: 0 (0.0%)	Grade 1: 4 (20.0%) Grade 2: 10 (50.0%) Grade 3: 4 (20.0%) Grade 4: 2 (10.0%)	

For continuous variables, the data are presented as mean ± SD and median (min–max). Categorical variables are presented as frequency (percentage). Data were analyzed using a one-way analysis of variance. Abbreviations: BMI, body mass index; SD, standard deviation; KL, Kellgren and Lawrence; 3′-SL, 3′-sialyllactose.

**Table 2 nutrients-16-03410-t002:** Adverse events according to the groups over the total study period.

	Placebo (*n* = 20)	200 mg 3′-Sialyllactose (*n* = 20)	600 mg 3′-Sialyllactose (*n* = 20)
Nausea	1	1	
Weight gain	1		
Heartburn	1		1
Indigestion	1		
Swelling	2	2	
Itching			1
Foamy urine	1		

**Table 3 nutrients-16-03410-t003:** VAS scores.

	Placebo	200 mg 3′-SL	600 mg 3′-SL	*p*-Value
Baseline	(*n* = 20)	(*n* = 20)	(*n* = 20)	
Mean ± SD Median (min–max)	34.00 ± 17.89 30.0 (10.0–70.0)	36.00 ± 16.98 30.0 (10.0–70.0)	40.50 ± 17.61 40.0 (20.0–90.0)	
6 weeks	(*n* = 20)	(*n* = 20)	(*n* = 18)	
Mean ± SD Median (min–max)	33.50 ± 14.24 30.0 (10.0–70.0)	24.00 ± 12.73 20.0 (10.0–50.0)	30.28 ± 17.70 30.0 (0.0–60.0)	
12 weeks	(*n* = 20)	(*n* = 18)	(*n* = 18)	
Mean ± SD Median (min–max)	23.06 ± 12.02 20.0 (10.0–50.0)	18.00 ± 11.05 ** 20.0 (0.0–40.0)	22.78 ± 17.76 * 20.0 (0.0–50.0)	** 0.0053, * 0.0115

For continuous variables, the data are presented as mean ± SD and median (min–max). Data were analyzed using a two-way analysis of variance followed by Tukey’s multiple comparisons test, * *p* < 0.05 and ** *p* < 0.01 vs. baseline. VAS, visual analog scale; 3′-SL, 3′-sialyllactose; SD, standard deviation.

**Table 4 nutrients-16-03410-t004:** KWOMAC scores.

	Placebo	200 mg 3′-SL	600 mg 3′-SL	*p*-Value
Baseline	(*n* = 20)	(*n* = 20)	(*n* = 20)	
Mean ± SD Median (min–max)	44.45 ± 8.64 43.5 (31.0–59.0)	44.55 ± 8.73 41.5 (32.0–60.0)	44.80 ± 14.28 41.0 (31.0–94.0)	
6 weeks	(*n* = 20)	(*n* = 20)	(*n* = 18)	
Mean ± SD Median (min–max)	43.45 ± 12.61 41.0 (27.00–76.0)	37.45 ± 9.97 35.0 (20.0–62.0)	34.50 ± 10.69 32.0 (19.0–53.0)	
12 weeks	(*n* = 20)	(*n* = 18)	(*n* = 18)	
Mean ± SD Median (min–max)	36.83 ± 11.38 32.0 (20.0–61.0)	34.55 ± 12.80 * 31.0 (21.0–66.0)	31.17 ± 7.76 ** 29.0 (21.0–53.0)	* 0.0326, ** 0.0018

For continuous variables, the data are presented as mean ± SD and median (min–max). Data were analyzed using a two-way analysis of variance followed by Tukey’s multiple comparisons test, * *p* < 0.05 and ** *p* < 0.01 vs. baseline. Abbreviations: KWOMAC, the Korean Western Ontario and McMaster Universities Osteoarthritis Index; 3′-SL, 3′-sialyllactose; SD, standard deviation.

## Data Availability

All data generated or analysed during this study are included in this published article and its Appendix A.

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
