# Peer review of "Effects of 3′-Sialyllactose on Symptom Improvement in Patients with Knee Osteoarthritis: A Randomized Pilot Study"

_nutrients, 2024, doi:10.3390/nu16193410_

Round 1

Reviewer 1 Report

Comments and Suggestions for Authors

The authors investigated the effect of 3'-SL on pain and physical function in patients with knee osteoarthritis. The manuscript is clear and presented in a well-structured manner. References cited are mainly recent publications and are relevant. Figures and tables are appropriate. Conclusions are consistent with the evidence presented. The authors should clarify a few points:

1. The authors should explain to what extent their article is new in the area of knowledge discussed.

2. I also suggest discussing the efficacy of 3′-sialyllactose in delivery injections.

3.  The authors should explain why they used 3′-SL at 200 and 600 mg doses.

4. Line 146-149. The text indicates the part of the review that recommends specific corrections to the manuscript.

Author Response

For Brief Report

Response to Reviewer 1 Comments

1. Summary

2. Questions for General Evaluation

Reviewer’s Evaluation

Response and Revisions

Does the introduction provide sufficient background and include all relevant references?

Can be improved

Is the research design appropriate?

Can be improved

Are the methods adequately described?

Yes

Are the results clearly presented?

Yes

Are the conclusions supported by the results?

Yes

3. Point-by-point response to Comments and Suggestions for Authors

The authors investigated the effect of 3'-SL on pain and physical function in patients with knee osteoarthritis. The manuscript is clear and presented in a well-structured manner. References cited are mainly recent publications and are relevant. Figures and tables are appropriate. Conclusions are consistent with the evidence presented. The authors should clarify a few points:

Comments 1: The authors should explain to what extent their article is new in the area of knowledge discussed.

Response 1: Thank you for pointing this out. We agree with this comment.

→ In vitro and in vivo studies have been shown that 3’-SL has improving effects against OA symptoms. Additionally, the safety of 3’-SL in a human pilot study has been reported. However, effect of 3’-SL on human OA symptoms has not been reported previously. Therefore, I’d like to highlight these as new facts in this field.

I have added the sentences in conclusion section as bellows;

This study is the first time to investigate the effects of 3’-SL on human OA symptoms. It is valuable and can be considered a new discovery in this field.

Page 10 / Section 5 / Line 343-345

Comments 2: I also suggest discussing the efficacy of 3′-sialyllactose in delivery injections.

Response 2: Thank you for pointing this out. We agree with this comment.

→ I have added the discussion about the efficacy of 3’-SL in delivery injection as bellows;

Since in vivo study of 3’-SL on OA (mouse, minipig) used oral ingestion to treat 3’-SL, oral route of administration was chosen for this pilot clinical trial. However, topical/transdermal administration, along with oral administration, are the most widely used routes of administration in the treatment of OA [40]. Oral administration has been reported to cause problems such as low bioavailability, perforation of the digestive tract, liver, and kidneys, and gastrointestinal injury [40]. Moreover, intra-articular injected 3’-SL in minipig model of rheumatoid arthritis was shown to have a therapeutic effect [41]. Administration injections also would be useful in the future to test the more efficient and rapid effects of 3'-SL. 

Page 10 / Section 4 / Line 330-338

Comments 3: The authors should explain why they used 3′-SL at 200 and 600 mg doses.

 Response 3: Thank you for pointing this out. We agree with this comment.

→ We selected the dose of 3’-SL based on in vivo studies in mouse and minipig OA models. In minipig model, 200 mg and 400 mg/head of 3’-SL showed therapeutic effects on OA symptoms. In a mouse model, the effects of 3'-SL were evaluated using 10, 50, and 100 mg/kg of 3'-SL, and 50 and 100 mg/kg of 3'-SL improved OA symptoms. 100 mg/kg in mice is equivalent to 8 mg/kg in humans (480 mg/person). We thought that a dose of 200 mg or more would be effective in humans. Therefore, 200 mg was selected as the low dose. 

Furthermore, 600 mg was selected as the high dose, considering the dose used in the mouse study (100 mg/kg (480 mg/person)).

I have added the sentence about the dose of 3’-SL in Materials and Method section as bellows;

Dosages were selected based on in vivo studies in mouse and minipig OA models [18,19].

Page 2 / Section 2.1. / Line 76-77

Comments 4: Line 146-149. The text indicates the part of the review that recommends specific corrections to the manuscript.

Response 4: Thank you for pointing this out. We agree with this comment.

→ I have deleted these sentences from manuscript.

Lastly, I would like to thank you for your time and effort in helping me to improve the quality of this paper. Thank you for your comments, and I hope that my revisions and modifications will meet your expectations.

Reviewer 2 Report

Comments and Suggestions for Authors

My comments are as follows:

Introduction on OA should be improved. OA is a whole joint disease, involving all joint tissues. It is characterized by subchondral bone remodeling, meniscal degeneration, inflammation and fibrosis of both infrapatellar fat pad and synovial membrane (doi: 10.1007/s11420-011-9248-6 etc). Moreover, infrapatellar fat pad and synovial membrane are involved in OA pain (DOI10.3389/fcell.2022.886604 etc).

Lines 29-30: this sentence needs to be modified. OA is a complex disease and obesity, ageing etc have an impact not only on cartilage but also on other joint tissues (such as infrapatellar fat pad). It should be simply stated that obesity, aging, previous injury etc are risk factors for OA.

Lines 33-34: Authors forget infrapatellar fat pad fibrosis and inflammation.

Lines 46-48: references should be added.

Line 60: authors enrolled patients between August 2019 and March 2020 and the patients were followed for 12 weeks. Thus, the study should be ended in March 2021. Why did the authors wait 3 years before submitting the study?

Line 62: a reference for KWOMAC should be provided.

Lines 63-65: how did authors select these doses?

Line 81: WOMAC or KWOMAC?

Line 84: a reference for ACR criteria should be added.

Section 2.4: could the authors provide more details about 3’-SL?

A priori sample size should be calculated.

Lines 145-149: this part should be deleted.

It should be clarified how randomization was performed.

Lines 152-155: it seems that authors excluded from the study three patients that did not obtain good results with the treatment. Clarification is needed.

Table 1: it is unclear if there are differences between the groups regarding KL.

Figure 1: authors need to check the figures. For example, baseline of 3’-SL 600mg reported in figure 1a is different compared to baseline 3’-SL 600 mg reported in figure 1b. Why?

Figure 2: authors should check as figure 2a does not correspond to figure 2b. I mean that baseline 3’sl 600 mg reported in a is different compared to that reported in b. Again, why?

Lines 244-245: Patients in the 3′-SL 600-mg group had 244 higher VAS scores at baseline than those in the other groups. This is not clear. No differences were detected by authors regarding VAS baseline in the different groups. Authors should check and correct.

Genus and species names should be written in italics.

Lines 270-273: I suggest to discuss systematic reviews with meta-analysis on this topic.

Lines 289-291: author forget that OA is characterized by biomechanical and fibrotic changes of the tissues.  

Lines 295-299 should be moved after line 288.

Author Response

For Brief Report

Response to Reviewer 2 Comments

1. Summary

2. Questions for General Evaluation

Reviewer’s Evaluation

Response and Revisions

Does the introduction provide sufficient background and include all relevant references?

Can be improved

Is the research design appropriate?

Can be improved

Are the methods adequately described?

Can be improved

Are the results clearly presented?

Can be improved s

Are the conclusions supported by the results?

Can be improved

3. Point-by-point response to Comments and Suggestions for Authors

Comment 1: Introduction on OA should be improved. OA is a whole joint disease, involving all joint tissues. It is characterized by subchondral bone remodeling, meniscal degeneration, inflammation and fibrosis of both infrapatellar fat pad and synovial membrane (doi: 10.1007/s11420-011-9248-6 etc). Moreover, infrapatellar fat pad and synovial membrane are involved in OA pain (DOI10.3389/fcell.2022.886604 etc).

Response 1: Thank you for pointing this out. We agree with this comment.

→ We have improved introduction which includes the contents of OA physiology, risk factors, and symptoms as bellows;

Knee Osteoarthritis (OA) is multifactorial and degenerative whole joint disease that cause damage to the cartilage and surrounding tissues such as subchondral bone, meniscal, ligaments, capsule, and synovium [1]. It is characterized by breakdown of the cartilage, subchondral bone remodelling, deterioration of tendons and ligaments, meniscal degeneration, inflammation and fibrosis of both infrapatellar fat pad and synovial membrane [1,2]. OA is the most common type of arthritis and known to be caused by several risk factors such as musculoskeletal aging, obesity, joint injury, anatomical factors, and excessive joint use [3]. As the damage of cartilage and joint tissue including infrapatellar fat pad and the adjacent synovial membrane, progresses, several symptoms of OA such as pain, stiffness, swelling near joint, loss of motion, and subsequent functional disability occurs [4]

Page 1 / Section 1 / Line 28-38

Comment 2: Lines 29-30: this sentence needs to be modified. OA is a complex disease and obesity, ageing etc have an impact not only on cartilage but also on other joint tissues (such as infrapatellar fat pad). It should be simply stated that obesity, aging, previous injury etc are risk factors for OA.

Response 2: Thank you for pointing this out. We agree with this comment.

→ We have stated that the several risk factors are correlated with OA progression as bellows;

OA is the most common type of arthritis and known to be caused by several risk factors such as musculoskeletal aging, obesity, joint injury, anatomical factors, and excessive joint use [3]. As the damage of cartilage and joint tissue including infrapatellar fat pad and the adjacent synovial membrane, progresses, several symptoms of OA such as pain, stiffness, swelling near joint, loss of motion, and subsequent functional disability occurs [4].

Page 1 / Section 1 / Line 33-38

Comment 3: Lines 33-34: Authors forget infrapatellar fat pad fibrosis and inflammation.

Response 3: Thank you for pointing this out. We agree with this comment.

→ We have considered about infrapatellar fat pad fibrosis and inflammation and revised the introduction of OA as bellows;

Knee Osteoarthritis (OA) is multifactorial and degenerative whole joint disease that cause damage to the cartilage and surrounding tissues such as subchondral bone, meniscal, ligaments, capsule, and synovium [1]. It is characterized by breakdown of the cartilage, subchondral bone remodelling, deterioration of tendons and ligaments, meniscal degeneration, inflammation and fibrosis of both infrapatellar fat pad and synovial membrane [1,2]. OA is the most common type of arthritis and known to be caused by several risk factors such as musculoskeletal aging, obesity, joint injury, anatomical factors, and excessive joint use [3]. As the damage of cartilage and joint tissue including infrapatellar fat pad and the adjacent synovial membrane, progresses, several symptoms of OA such as pain, stiffness, swelling near joint, loss of motion, and subsequent functional disability occurs [4]

Page 1 / Section 1 / Line 28-38

Comment 4: Lines 46-48: references should be added.

Response 4: Thank you for pointing this out. We agree with this comment.

→ We have added the references.

Page 2 / Section 1 / Line 49-50

Comment 5: Line 60: authors enrolled patients between August 2019 and March 2020 and the patients were followed for 12 weeks. Thus, the study should be ended in March 2021. Why did the authors wait 3 years before submitting the study?

Response 5: Thank you for pointing this out. We agree with this comment.

→ We got positive results on VAS and KWOMAC test in this pilot clinical study. However, numbers of patients and site were too small to market 3’-SL. Therefore, we have considered clinical trial in more numbers of patients at several clinical sites and planned to report these study results. However, the clinical trials conducted by the CRO took longer than initially expected, and various problems arose within the company, so the study was discontinued.

And we ended up reporting the first pilot clinical trial study now.

Comment 6: Line 62: a reference for KWOMAC should be provided.

Response 6: Thank you for pointing this out. We agree with this comment.

→ We have added a reference for KWOMAC.

Page 2 / Section 2.1. / Line 64

Comment 7: Lines 63-65: how did authors select these doses?

Response 7: Thank you for pointing this out. We agree with this comment.

→ We selected the dose of 3’-SL based on in vivo studies in mouse and minipig OA models. In minipig model, 200 mg and 400 mg/head of 3’-SL showed therapeutic effects on OA symptoms. In a mouse model, the effects of 3'-SL were evaluated using 10, 50, and 100 mg/kg of 3'-SL, and 50 and 100 mg/kg of 3'-SL improved OA symptoms. 100 mg/kg in mice is equivalent to 8 mg/kg in humans (480 mg/person). We thought that a dose of 200 mg or more would be effective in humans. Therefore, 200 mg was selected as the low dose. 

Furthermore, 600 mg was selected as the high dose, considering the dose used in the mouse study (100 mg/kg (480 mg/person)).

I have added the sentence about the dose of 3’-SL in Materials and Method section as bellows;

Dosages were selected based on in vivo studies in mouse and minipig OA models [18,19].

Page 2 / Section 2.1. / Line 76-77

Comment 8: Line 81: WOMAC or KWOMAC?

Response 8: Thank you for pointing this out. We agree with this comment.

→ We used KWOMAC, which is comprehensible, reliable, valid, and responsive instruments to measure outcome in patients with knee OA in Korea, and their psychometric properties are comparable with those of the original versions (doi:10.1053/joca.2001.0471). We have added the reference in Material and Methods section.

Page 2 / Section 2.1. / Line 64

Comment 9: Line 84: a reference for ACR criteria should be added.

Response 9: Thank you for pointing this out. We agree with this comment.

→ We have added a reference for ACR criteria.

[Altman, R.; Asch, E.; Bloch, D.; Bole, G.; Borenstein, D.; Brandt, K.; Christy, W.; Cooke, T.D.; Greenwald, R.; Hochberg, M.; Howell, D.; Kaplan, D.; Koopman, W.; Longley, S.; Mankin, H.; McShane, D.J.; Medsger, T.; Meenan, R.; Mikkelsen, W.; Moskowitz, R. Development of criteria for the classification and reporting of osteoarthritis: Classification of osteoarthritis of the knee. Arthritis & Rheumatism 1986, 29, 1039-1049.]

Page 3 / Section 2.3. / Line 99

Comment 10: Section 2.4: could the authors provide more details about 3’-SL?

Response 10: Thank you for pointing this out. We agree with this comment.

→ We have added the explanations the way to produce 3’-SL as bellows;

3’-SL was produced using the one pot reaction system invented by GeneChem and dissolved in water. Briefly, substrates and enzymes (cytidylate kinase (CMK), polyphospate kinase (PPK), CMP-NeuAc synthetase (NEU), N-acetyl-D-glucosamine-2-epimerase (NANE), NeuAc aldolase (NAN), and α2,3-sialyltranferase (α2,3STN)) were mixed and reacted in a one-pot reactor.

Page 3 / Section 2.4. / Line 126-130

Comment 11: A priori sample size should be calculated.

Response 11: Thank you for pointing this out. We agree with this comment.

→ This study was a pilot clinical study and assessed the feasibility of whether a larger trial should be initiated. It has been reported that sample size is often not calculated in pilot studies and some studies recommend over 30 samples per group while some suggest 12 per group (doi: 10.4097/kjae.2017.70.6.601).

We have added the sentence about sample size calculations as bellows;

This study is pilot clinical study and a formal statistical power analysis to determine the study sample size was not completed.

Page 4 / Section 2.6. / Line 160-161

Comment 12: Lines 145-149: this part should be deleted.

Response 12: Thank you for pointing this out. We agree with this comment.

→ We have deleted these sentences.

Page 4 / Section 2.6. / Line 162-165

Comment 13: It should be clarified how randomization was performed.

Response 13: Thank you for pointing this out. We agree with this comment.

→ We have used block randomization method. We have added the explanations of randomization as bellows;

For patients who meet all exclusion criteria, random assignment numbers are randomly assigned to the experimental group or control group in a 1:1 ratio in the order registered by the investigator. A random assignment random number table is generated by a statistician independent of this clinical trial using statistical software (Vserion 9.4, SAS® Institute, Cary, NC, USA) using the block randomization and a random assignment number of sufficient size is issued considering the pre-designated block size. A random assignment envelope is made with the issued random assignment number, and the random assignment envelope is delivered to the unblind investigator of the institution before subject registration.

Page 2 / Section 2.1. / Line 66-74

Comment 14: Lines 152-155: it seems that authors excluded from the study three patients that did not obtain good results with the treatment. Clarification is needed.

Response 14: Thank you for pointing this out. We agree with this comment.

→ Two patients on 3′-SL 600 mg were lost to follow-up by 6 weeks and two patients on placebo were lost to follow-up by 12 weeks. These patients were excluded because efficacy could not be assessed, not because they did not have a good outcome.

Page 4 / Section3.1. / Line 169-170

Comment 15: Table 1: it is unclear if there are differences between the groups regarding KL.

Response 15: Thank you for pointing this out. We agree with this comment.

→ The number of patients with KL grade 1 was 4, 3, and 4 for placebo, 200 mg, and 600 mg, respectively. And the number of patients with KL grade >1 was 16, 17, and 16 for placebo, 200 mg, and 600 mg, respectively. There are no significant differences between the groups regarding KL.

Comment 16: Figure 1: authors need to check the figures. For example, baseline of 3’-SL 600mg reported in figure 1a is different compared to baseline 3’-SL 600 mg reported in figure 1b. Why?

Response 16: Thank you for pointing this out. We agree with this comment.

→ We have checked figures carefully and carried out statistical analysis again. The results of the statistical analyses have been modified and the revised figures are presented in the Results section. Moreover, we have revised the discussion in response to the revied results.

Page 5 / Section3.3.1. / Line 195-199

Page 5 /Section3.3.1. / Line 206-209

Page 8 /Section 4 / Line 255-257

Comment 17: Figure 2: authors should check as figure 2a does not correspond to figure 2b. I mean that baseline 3’sl 600 mg reported in a is different compared to that reported in b. Again, why?

Response 17: Thank you for pointing this out. We agree with this comment.

→ We have checked figures carefully and carried out statistical analysis again. The results of the statistical analyses have been modified and the revised figures are presented in the Results section. Moreover, we have revised the discussion in response to the revied results.

Page 6 / Section3.3.2. / Line 225-228

Page 7 / Section3.3.2. / Line 235-238

Page 8 / Section 4 / Line 266-280

Comment 18: Lines 244-245: Patients in the 3′-SL 600-mg group had higher VAS scores at baseline than those in the other groups. This is not clear. No differences were detected by authors regarding VAS baseline in the different groups. Authors should check and correct.

Response 18: Thank you for pointing this out. We agree with this comment.

→ As you pointed out, no significant differences were detected regarding VAS baseline among three groups. Therefore, I have deleted this sentence in discussion section.

Comment 19: Genus and species names should be written in italics.

Response 19: Thank you for pointing this out. We agree with this comment.

→ We have written genus and species names in italics.

Treatment of Boswellia serrata, ginger extracts, and Harpagophytum procumbens reduced symptoms of knee OA in clinical trials [31].

Page 9 / Section 4 / Line 286-287

Comment 20: Lines 270-273: I suggest to discuss systematic reviews with meta-analysis on this topic.

Response 20: Thank you for pointing this out. We agree with this comment.

→ We have discussed about systematic reviews with meta-analysis on OA treatment in Discussion section as bellows;

According to the network meta-analysis of treatments for knee OA through PubMed, Embase, and Cochrane Library electronic databases from the inception to October 2018, hyaluronic acid combined with platelet-rich plasma showed efficiency in improving physiological function and stiffness, and total score in WOMAC, while platelet-rich plasma reduced pain in WOMAC [35]. Another network meta-analysis of intervention for OA by Cochrane reviews showed diet/weight loss and surgery has less effect than corticosteroids, herbs, mind and body exercises, orthotics, passive treatment, and regenerative medicine on pain [36].

Page 9 / Section 4 / Line 288-295

Comment 21: Lines 289-291: author forget that OA is characterized by biomechanical and fibrotic changes of the tissues.  

Response 21: Thank you for pointing this out. We agree with this comment.

→ I have added the discussion about biochemical and fibrotic changes of the tissues in Discussion section as bellow;

Although the exact mechanism remains unclear, OA has been associated with multifactorial events such as inflammation, MMP-induced collagen degradation, hyperactivated catabolic activity, and oxidative stress responses, joint fibrosis, and biomechanical changes [16-18, 29, 38, 39]. Studies have evaluated the action mechanism of 3′-SL in in vitro and in vivo OA models and showed that 3′-SL inhibits cartilage degradation, inflammation, oxidative stress, cell apoptosis, and promoted cartilage regeneration [17-19]. These mechanisms of 3′-SL may be influenced to relieve pain and enhance physical function. Moreover, it would be worthwhile to examine whether 3′-SL inhibits the excessive secretion and deposition of extracellular matrix proteins that induce excessive fibrosis. Tissue fibrosis induces joint stiffness and pain in OA [38]. As is known that OA is a disease of the whole joint, biomechanical changes also affect the pathogenesis and progression of OA [39]. Effects of 3’-SL on biomechanical tissue characteristics such as muscle atrophy, bone density, subchondral bone sclerosis also should be determined.

Page 9-10 / Section 4 / Line 316-328

Comment 22: Lines 295-299 should be moved after line 288.

Response 22: Thank you for pointing this out. We agree with this comment.

→ I have moved these sentences.

Page 9 / Section 4 / Line 312-315

Lastly, I would like to thank you for your time and effort in helping me to improve the quality of this paper. Thank you for your comments, and I hope that my revisions and modifications will meet your expectations.

Round 2

Reviewer 2 Report

Comments and Suggestions for Authors

No additional comments.